# MULTIVARIATE TIME-SERIES FORECASTING WITH SPACE: SERIES PREDICTION AUGMENTED BY CAUSALITY ESTIMATION

## ABSTRACT

The analysis of multivariate time series (MTS) presents a complex yet crucial task with substantial applications in areas such as weather forecasting, policy formulation, and stock market prediction. It is important to highlight three key characteristics of MTS that contribute to the challenging and multifaceted nature of their analysis: (i) their interrelationships are represented through causal relationships rather than mere similarities; (ii) they convey information across multiple independent factors; and (iii) their dynamics often arise from inherent temporal dependencies. While conventional time series analysis frameworks often fail to capture one or more of these aspects, resulting in incomplete or even misleading conclusions, we propose an end-to-end trainable **S**eries **P**rediction model **A**ugmented by **C**ausality **E**stimation (SPACE) to address these limitations. This model effectively incorporates temporal dependencies and causal relationships, featuring a temporal embedding and a transfer entropy-based Cross-TE module designed to enhance predictions through causality-augmented mechanisms. Experiments demonstrate that SPACE achieves state-of-the-art results on challenging real-world time series prediction tasks, showing its effectiveness and versatility. Code is available at `https://anonymous.4open.science/r/SPACE-D448/`.

## 1 INTRODUCTION

Time series forecasting (TSF) is an inherently difficult problem. A large part of this is due to the overall structural complexity of time series information. This is especially true for time series that reflect real-world data, for example those that record usage statistics of the electrical grid, local temperature variations over a specific time window, or market values of stocks. On the one hand, it is understood that the diverse length scales and temporal dynamics underlying each of these systems is the main reason for their richness and insight; conversely, they are also the reason for the intractability of many real-world time series. In the examples listed above, the frequency and distribution of electrical grid value fluctuations are greatly influenced by temporal and seasonal trends, sudden changes in demand (e.g., during a popular sporting event, when a large number of users tune in to the sports broadcast; or during an unexpected heat wave or cold snap), and the spatial location and role of possible malfunctioning grid nodes, which affect supply to a subset of users. Similarly, the changes in stock values also governed by a multitude of factors, both overt and latent, and these often interact in a variable and nonlinear manner. The analysis of these examples is most naturally expressed in the language of causality.

In contrast, most current TSF frameworks Liu et al. (2023); Wu et al. (2021); Oreshkin et al. (2020) approach the problem of forecasting from the perspective of *similarity* and *multivariate dependencies*; in other words, at the base level, these approaches are mainly focused on learning the correlative weights between time series. This is especially the case for SOTA attention-based models Liu et al. (2023); Chen et al. (2024); Wang et al. (2024b), as these models focus specifically on learning attention weights between different preprocessed time series. All these different approaches excel at capturing different facets of complex time series, but they are all predicated upon the same idea of correlation and similarities. This presents a singularly one-dimensional view of conventional time series analysis and hence hampers comprehensive and in-depth understanding of this information.

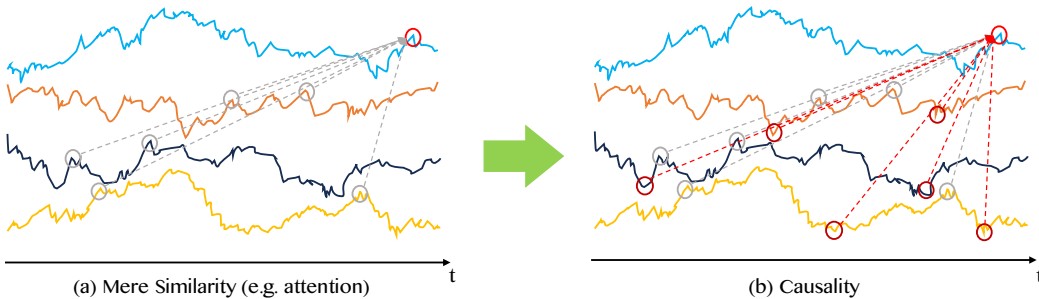

Figure 1: (a) Models that make use of correlative weights only focus more on points that are similar to the target, tending to ignore different pattern. (b) Causality is capable of capturing dissimilar information that is favourable to prediction tasks in addition to similar one.

As mentioned previously, this unsatisfactory situation with regards with current SOTA TSF models can traced to a key omission: the neglect of *causal* relationships. They play an essential role in understanding the causes and effects influencing temporal behavior. They have been shown Moraffah et al. (2021); Runge et al. (2023) to provide essential perspectives into the temporal dimension and, therefore, should contain invaluable insights for a comprehensive understanding of time series. In fact, the idea of precedence in a time series and causality are inextricably linked together and it is misleading to analyze temporal information without considering causal relations Wikipedia contributors (2024). Although there is a significant body of work on TSF models that do Kong et al. (2024); Nichani et al. (2024); Börjesson & Singull (2020); Nauta et al. (2019); Cheng et al. (2024); Löwe et al. (2022); Chikahara & Fujino (2018); Dhaou et al. (2021) take causal information into account, their focus is usually misplaced, in that their contributions tend to either *focus exclusively on temporal causal relationships*, or *treat causality as another aspect of time series to be studied*. The first type of models Börjesson & Singull (2020); Nauta et al. (2019); Cheng et al. (2024) mainly emphasizes analysis of time series from the sole perspective of causality, while discarding other, equally important, features. The other type of approach Kong et al. (2024) looks at causality as merely one facet of time series data, and hence is susceptible to de-emphasizing its importance.

We take an integrative approach which promotes the causal aspect to a level which places it on the same footing as other time series features; in other words, we propose an integrative approach that takes both *temporal dependencies* and *causal* information into account. We believe that the intractability of naturally-occurring time series can be understood within the all-encompassing context of *causality*. By starting from an intuitive understanding of causality as the study of *cause* and *effect* relationships, we further leverage causality as the guiding principle according to which we can comprehend the above-mentioned complexities of real-world time series. In our framework, we extract temporal causal information in the form of embedding vectors, which we integrate via a spatial causal module to facilitate downstream forecasting tasks. For this purpose, we propose *SPACE*, a model which is able to comprehensively take both these aspects of time series into account. SPACE is inspired by the modular design of the original transformer and graph neural networks, augmented by several main elements:

- A *Sequence Enhancer*, whose role is to compute correlational coefficients between the patched and projected raw data from the embedding step, using an attention mechanism;

- A *Cross TE* module which gets causality information by computing transfer entropy (TE) self-causal relationships between time series using fast-pTE algorithm promoted by us;

- A *Causal Graph Neural Network* (CGNN) which integrates information by taking causality matrix from cross-TE modules as its adjacency matrix, in order to present a causally-consolidated embedding vector for the final downstream tasks;

We integrate the above-mentioned modules in a conventional attention framework, which enables our workflow to function as a drop-in replacement for attention modules. In short, our contribution in this work are as follows:

- We show that for a large class of time series data, correlative information is insufficient for a comprehensive understanding; instead, a causative view is much more informative in comparison;

- Based on the observations regarding correlative vs causative observations, we design two novel modules which are formulated to take these additional insights into account: the *Cross TE* layers, and *Causality-based* Graph Network;

- In order to reduce time and space complexity which is relatively high for deep learning tasks in original pseudo transfer entropy calculation, we propose a faster algorithm, *Fast-pTE*. It not only reduces the complexity quadratically, from $O(d^2T)$ to $O(dT)$, where $d$ is the hidden size of input data, and $T$ is the number of time states, but also promotes higher performance compared to the original one.

- We show, via numerical experimentation on several datasets, that by explicitly taking causative information into account, our model is able to outperform several SOTA attention-based time series forecasting models.

## 2 RELATED WORK

### 2.1 TRANSFORMER-BASED TIME SERIES ANALYSIS

Time series analysis techniques are very well-developed. State-of-the-art time series analysis models incorporate the most recent advances in sequential data analysis, including various modifications to the basic transformer architecture Vaswani et al. (2017). Liu et al. (2023) inverts the traditional time series embedding for attention models, such that they now emphasize the attentive correlation between variates, taking into account the full set of timepoints for each variate into account. Zhang & Yan (2023) utilizes cross-dimensional dependencies between related variates to enhance time-series prediction accuracy. Nie et al. (2023) further emphasizes the importance and advantages of patching for time series forecasting, and interprets each patch as semantic, making it to provide a new perspective on its function. Wu et al. (2021) is an approach which actively decomposes the time series into logically coherent substructures (long-term trends, intermediate scale fluctuations, etc) and uses these simpler substructures to enhance predictive power of the model. Lin et al. (2023) includes learnable placeholders in the input embedding to the transformer encoder, thus achieving increased accuracy and reduced model complexity. Zhang et al. (2024) takes multiscale data into account by extending patch-based TS transformers with attention mechanisms that learn multiresolution representations; Finally, Wang et al. (2024b) incorporates exogenous variables into the learning process, hence taking into account the effect of such variables on the dynamics of the time series process. unlike most attention-based models described here, Wang et al. (2024b) is able to explicitly reason about dynamics of a specific time series contingent on possible influencing factors.

### 2.2 GRANGER CAUSALITY VIA TRANSFER ENTROPY

In recent years, causality has become a well-studied and essential component of time-series analysis. The original proposal, by Granger Granger (1969), was actually meant to analyze "precedence", in the sense that, for two series $X$ and $Y$, $Y$ is said to be *forecast* by $X$, if there exists a Granger-causal relationship between them. In most applications involving causal relationships, Granger-causality is inferred from the *transfer entropy* Schreiber (2000), which is a non-parameteric statistic originating from the physics literature. It is an information theoretic measure that quantifies the amount of information transfer between two random processes Hlaváčková-Schindler et al. (2007). It has been shown Barnett et al. (2009) that Granger causality and transfer entropy are the same for a stream of normal-distributed random variables. Although conditioning on the distribution restricts the applicability of transfer entropy as a causality surrogate, it has been widely utilized in this context because of its ease of computation.

### 2.3 FINANCIAL TIME SERIES PREDICTION

There is a substantial amount of works related to the prediction or *forecasting* of financial time series; in this section we consider, in particular, those applying deep learning methods. Conventional deep approaches include long short-term memory Hochreiter & Schmidhuber (1997), convolutional

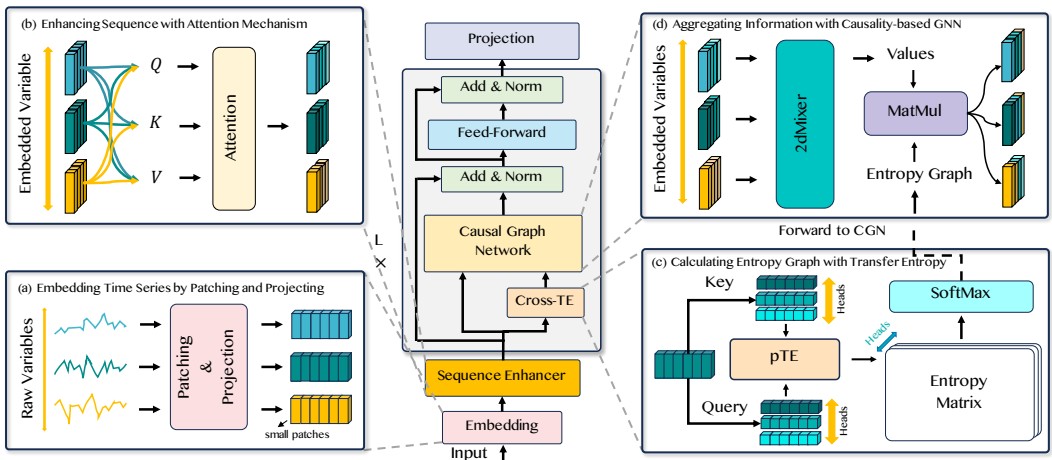

Figure 2: Overall structure of SPACE. (a) Raw variables are patchified into 2D series, and are subsequently projected as embedded tokens. (b) A Sequence Enhancer module is used to preprocess data. (c) Embedded variables are segmented into heads, followed by the application of pseudo transfer entropy (pTE) to compute the causal relationships between segmented sequences. (d) A causality-based graph neural network is applied to capture dependencies among variables, in which there is a 2dMixer to mix information within and across patches before aggregating from without.

neural networks Bai et al. (2018), or hybrid approaches combining several deep modules. More recent approaches have included transformer-based approaches. Several models mentioned in the section on transformer-based approaches have also been successfully applied to financial time-series forecasting. Among these, Oreshkin et al. (2020) have been shown to give accurate predictions on the stock S&P index; this model consists of a setup where a both forecast and backcast expansion coefficients are learned using FC layers, and these learned values are used to construct a predictor for both the backcast and forecast values. Ding et al. (2021) is a recent attention-based model which aims to take the essential characteristics of stock series into account: information on multiple temporal scales, as well as hierarchical dependencies between series. They achieve this via a multiscale Gaussian prior, and orthogonally regularized attention heads.

## 3 METHODOLOGY

In time series forecasting, it is common to encounter with occasion when multivariate data is used for prediction. Given historical observation with N dimension $\boldsymbol{X} = \{\boldsymbol{x}_1, \boldsymbol{x}_2, ..., \boldsymbol{x}_L\} \in \mathbb{R}^{N \times L}$ where L is the look back window length, we predict future T time steps $\boldsymbol{Y} = \{\boldsymbol{x}_{L+1}, \boldsymbol{x}_{L+2}, ..., \boldsymbol{x}_{L+T}\} \in \mathbb{R}^{N \times T}$. It is worth noting that there may be a causal relationship between time series of different dimensions in the same set of data, that is, if the time series $\boldsymbol{y}$ is caused by $\boldsymbol{x}$, then introducing $\boldsymbol{x}$ when predicting $\boldsymbol{y}$ will definitely help improve the accuracy of the model results.

### 3.1 STRUCTURE OVERVIEW

Our proposed model SPACE is illustrated in Figure 2, consisting of modules: *Preprocessor* which contains *Embedding* and *Sequence Enhancer*, *Cross TE*, *Causal Graph Neural Network*, and *Projector*.

#### 3.1.1 PREPROCESSOR

Before we start our discussion about embedding, we would like to review the two methods previously used by mainstream multivariate time series prediction models, and contrast these with our own approach towards time series embedding. The method is *cross-sectional*, i.e., all the data points occurring at the same time are turned into a column vector Wu et al. (2021); Zhou et al. (2021) for

embedding. The limitation of this method is obvious, as it only focuses on obtaining the dependencies in the time dimension, and the dependencies across the sequences are learnt only through embedding and subsequent linear projection, which limits the ability to help predict between different sequences while adding a lot of noise to the prediction of a single time series. The second method is *cross-temporal*, i.e., each individual time series is seen as a vector. The whole time series is either treated as a token for embedding Liu et al. (2023) or patchified Nie et al. (2023); Zhang & Yan (2023).

While the former approach absorbs the idea that linear projection is capable of learning cross-time dependencies alone without any help from other structure including attention mechanism Zeng et al. (2022); Li et al. (2023), in our point of view, the latter approach considers the properties of time series from the perspective of the time period, rather than being restricted to individual points in time, which can be utilized to lower the high uncertainty in single time point, improving overall performance of modules such as our *Cross-TE*. To illustrate, given raw data $\forall \boldsymbol{x}_i \in \boldsymbol{X}$, we first patchify them into a 2D tensor $\boldsymbol{h}_i \in \mathbb{R}^{P_N \times P_L}$, where $P_N$ is the number of patches and $P_L$ refers to patch length. The patchified data will then be mapped to latent space of dimension $d$ via trainable linear projection. After embedding, series are inputted into our Sequence Enhancer module, which serve as the method to share information and patterns from other patches, through which we believe can enhance a sequence for latter process since it can better utilize the characteristic of auto-regressivity in time series. In this module, we adopt merely a multi-head attention block, with input-token linear mapping with $\boldsymbol{W}_i$, $\boldsymbol{b}_i \in \mathbb{R}^{d \times d}$ and $\{\boldsymbol{Q}, \boldsymbol{K}, \boldsymbol{V}\} = \boldsymbol{H} \cdot \boldsymbol{W}_i + \boldsymbol{b}_i, i \in \{q, k, v\}$. Then through output linear mapping, subsequently followed by `LayerNorm` and residual connection, we get $\boldsymbol{H}$ as the output that should be forwarded to encoder layers. It is worth noting that with attention and residual connection, we not only enhance the series but also preserve its time steps information, which is of great importance for the computation of transfer entropy.

### 3.1.2 CROSS-TE

As stated in the introduction, compared to causative analysis, correlation is not the optimal way to extract information from time series data. In order to realize the computation of causal weights, we employ transfer entropy (TE) methods, which we detail in this section. As alluded to in the introduction, the TE is a measure of the directional information flow from one time series to another, quantifying the influence of one process on the future state of another. It can be defined as

$$T_{X \to Y} = \sum P(i_{n+1}, i_n^{(k)}, j_n^{(l)}) \log \frac{P(i_{n+1} \mid i_n^{(k)}, j_n^{(l)})}{P(i_{n+1} \mid i_n^{(k)})} \tag{1}$$

where $P(\cdot, \cdot, \cdot)$ and $P(\cdot \mid \cdot)$ represent joint probability and conditional probability respectively, $i_{n+1}$ is the state of process $X$ at time step $n+1$, $i_n^{(k)}$ and $j_n^{(l)}$ are shorthand notations that represents the states of $X$ and $Y$ the previous $k$ and $l$ time steps. The advantage of TE lies in its ability to model causal dependencies that are not limited to linear, making it suitable for our task. However, traditional computation method of TE can cause performance bottleneck since it is relatively expensive to compute. Therefore, an alternative, the pseudo transfer entropy (pTE) Silini & Masoller (2021b), which is cheaper in terms of computational overhead is applied instead. The pTE assumes that all the time series follow the normal distribution, which is an acceptable assumption for real-world time series.

To be specific, pTE from time series $\boldsymbol{y}$ to $\boldsymbol{x}$ can be given by the formula:

$$\text{pTE}_{x \to y} = \frac{1}{2} \log \left( \frac{|\Sigma(\boldsymbol{I}_t \oplus \boldsymbol{J}_t)| \cdot |\Sigma(\boldsymbol{i} \oplus \boldsymbol{I}_t)|}{|\Sigma(\boldsymbol{i} \oplus \boldsymbol{I}_t \oplus \boldsymbol{J}_t)| \cdot |\Sigma(\boldsymbol{I}_t)|} \right) \tag{2}$$

where $\boldsymbol{I}_t$ and $\boldsymbol{J}_t$ represent past observations of $\boldsymbol{y}$ and $\boldsymbol{x}$ respectively, $\boldsymbol{i}$ is the future value of $\boldsymbol{y}$, $\Sigma(\boldsymbol{A} \oplus \boldsymbol{B})$ is the covariance of matrix $\boldsymbol{A}$ concatenated with $\boldsymbol{B}$. Furthermore, the original algorithm for high dimensional pTE still retains its quadratic complexity in the hidden dimension. Hence, we introduce the *fast-pTE* algorithm, which flattens the series in the last two dimension before applying conventional pTE, which not only reduces the complexity quadratically, but also lowers loss in many datasets. Full details can be found in the Appendix.

While TE serves as an effective method for identifying causal relationships between time series, its application to individual series often limits the analysis to the causality present within the current

look-back window, thus failing to capture broader, global patterns. Pre-computing TE values offers a potential solution Duan et al. (2022), yet it overlooks the dynamic nature of causal relationships, which can fluctuate due to factors such as periodic behaviors or abrupt changes in the data. To address these limitations, we introduce the *Cross-TE* block, designed to dynamically learn and adapt to evolving causal dependencies while preserving memory of past relationships.

For the single head version of Cross-TE, suppose we have embedded time series $\boldsymbol{H} \in \mathbb{R}^{N \times PN \times d}$ where N denotes number of variables or dimensions of data, $P_N$ denotes number of patches in a single series, and $d$ is the hidden size of each patch. After projection: we get queries $\boldsymbol{Q}$ and keys $\boldsymbol{K}$, both of which have the same shape as $\boldsymbol{H}$. Then we can directly apply the fast-pTE formula to $\boldsymbol{Q}$ and $\boldsymbol{K}$ to get transfer entropy matrix $\boldsymbol{T} \in \mathbb{R}^{N_q \times N_k}$:

$$\boldsymbol{T} = \text{SoftMax}(\text{pTE}_{\boldsymbol{K} \rightarrow \boldsymbol{Q}}) \tag{3}$$

However, causality between time series can have various origins since there may be multiple factors at play in a real-world time series. For such a situation, a multi-headed attention is needed. In this situation, we divide $\boldsymbol{Q}$ and $\boldsymbol{K}$ in the last dimension, into $\boldsymbol{Q}_i$ and $\boldsymbol{K}_i \in \mathbb{R}^{N \times PN \times \frac{d}{h}}$ $(i = 1, 2, ..., h)$ where $h$ is the number of heads. Then T is simply a concatenation of $\boldsymbol{T}_i$:

$$\begin{aligned} \boldsymbol{T}_i &= \text{SoftMax}(\text{pTE}_{\boldsymbol{K}_i \rightarrow \boldsymbol{Q}_i}) \\ \boldsymbol{T} &= \text{Concat}(\boldsymbol{T}_1, \boldsymbol{T}_2, ..., \boldsymbol{T}_h) \end{aligned} \tag{4}$$

### 3.1.3 Causal Graph Neural Network

Information between series and dependencies across time plays a vital role in time series forecasting tasks. The question of how to perfectly aggregate this information, as well as how to take advantage of the temporal property, is of great importance. Therefore, we propose the *Causal Graph Neural Network* (CGNN) in order to take full account in both in cross-temporal and cross-dimensional situations.

Our transfer entropy matrix as computed in the Cross-TE module is a powerful tool for tackling problems in terms of aggregating information from others beyond a series itself, since causality reveal their essential relation. However, causality is a highly abstract concept, and a simple summation of values following linear mapping in transformer encoders will give misleading results. In addition, information among patches in each series needs to be shared for the same reason, for which a single linear mapping along hidden dimension is insufficient. These limitations underscores the need to apply graph neural networks to more accurately capture the intricate relationships among series.

Previous works that use transfer entropy in graph neural network simply adopt primitive graph neural networks, such as GCN and GIN, propagating information layer by layer through the graph structure embedded in the adjacency matrix to gradually integrate more global information Duan et al. (2022). They tend to calculate transfer entropy matrices before training models, which are then converted to sparse adjacency matrices by $\max(thresh, T_0)$ where $thresh$ is the minimum causality the model tend to consider, and $T_0$ is the entropy score. This method offers the advantage of simplifying calculations by uniformly and equivalently treating strongly causal sequences, but nevertheless falls short in distinguishing differences among them. So an all-pair message passing graph network with weights is called for. To simplify our description, we will only discuss the process in a single head. The transfer entropy score $T_0 \in \mathbb{R}^{N \times N}$ calculated by Cross-TE module will be normalized by SoftMax to $T$, and forwarded to CGNN. A function $f$, which is flexible to choose, will act on input data $H^0 \in \mathbb{R}^{N \times t \times d}$ from Sequence Enhancer, where $t$ and $d$ represent number of time state and hidden dimension respectively, then left-multiplied by $T$. These can be described by formulas below:

$$\boldsymbol{H}^{(k)} = \boldsymbol{T} \cdot f(\boldsymbol{H}^{k-1}) \tag{5}$$

where $k$ denotes the $k$-th layer in our GNN. Performance can vary if different $f$s are chosen, while we simply adopt a *2DMixer*, consisting of Patch and Time mixers. The former is a linear map that act within patch dimension, and the latter a non-linear function like MLP.

$$\begin{aligned} \boldsymbol{H}_{tmp} &= \texttt{Patch-Mixer}(\boldsymbol{H}^{k-1}) \\ \boldsymbol{H}_{tmp} &= \boldsymbol{H}_{tmp} \cdot \texttt{Permute}(0, 2, 1) \\ \boldsymbol{H}^k &= \texttt{Time-Mixer}(\boldsymbol{H}_{tmp} \cdot \texttt{Permute}(0, 2, 1)) \end{aligned} \tag{6}$$

This approach takes hidden dimension and time states of a series as different aspects, reducing computation cost while holding the same performance.

After the aforementioned processes, the outputs from the different heads of the *2DMixer* are concatenated and undergo a linear projection across the heads. Subsequently, they are added to $\boldsymbol{H}^0$ via a residual connection to preserve temporal information and enhance the efficiency of back propagation.

## 4 EXPERIMENTS

We conduct our experiment on 9 real-world datasets, including (1) **ETT** Zhou et al. (2021) contains 4 datasets with 7 sub-series of electricity data from July 2016 to July 2018 in it. ETTh1 and ETTh2 are two of the datasets recorded every hour, while ETTm1 and ETTm2 are two recorded every 15 minutes. (2) **Weather** Wu et al. (2021) contains 21 indicators of weather condition, such as air temperature and humidity, which are recorded every 10 minutes in 2020 from the Weather Station of the Max Planck Biogeochemistry Institute. (3) **Exchange-rate** Wu et al. (2021) is a dataset collecting daily exchange rates from 8 countries from 1990 to 2016. (4) We also provide the experiments on three financial index data. Detailed information on all datasets in use is available in the Appendix.

### 4.1 BASELINES AND SETUP

We compare our model by carefully choosing 8 well-acknowledged state-of-the-art models as our benchmark, including 3 linear-based methods: **DLinear** Zeng et al. (2022), **TiDE** Das et al. (2023), **RLinear** Li et al. (2023); 4 Transformer-based methods: **iTransformer** Liu et al. (2023), **PatchTST** Nie et al. (2023), **Crossformer** Zhang & Yan (2023), **Stationary** Liu et al. (2022); 1 TCN-based method: **TimesNet** Wu et al. (2023a). All of our experiments are conducted using PyTorch and executed on an NVIDIA RTX 4090 GPU. To ensure fair comparison, all model follow the same input length ($H = 96$) and prediction length ($F \in \{96, 192, 336, 720\}$). Parameters of competitive models follows the setting of Wu et al. (2023b) and Wang et al. (2024a) to eliminate influence caused by wrong parameter settings.

### 4.2 MAIN RESULTS AND DISCUSSION

The comprehensive forecasting results are listed in Table 1 with the best in red and second best in blue and underlined. SPACE shows the best performance in comparison with the baseline models across many real-world datasets and prediction length settings. Specifically, there are 69 first place and 10 second place rankings out of 90 comparison points. Hence we can conclude that the integration of causal information is essential to improving forecasting performance. On the public dataset, SPACE's performance exceeds those of the attention-based baselines by a considerable margin. The improvements in MSE and MAE for different prediction lengths $S$ are not linearly correlated with the baselines' performances, indicating a more fundamental departure in the design of SPACE, compared to the baselines' architectures, than a simple difference in depth or width of network architecture, which are all changes in *degree* rather than a change in *form*. This difference implies that the improvements we see in our results cannot be reproduced simply by modifying the degree complexity of the models. This departure is entirely due to the tight integration of *causal* modules with the conventional attention-based logic for time series forecasting.

In addition to the differences in computed metrics (MSE and MAE), results obtained on the real-world datasets show that causality enhances *interpretability* of time series forecasting as well. We illustrate this point via Fig. 3, which is a visualization of the attention key-queries adjacency matrix, as evaluated on the **Weather** dataset. This particular dataset contains numerous weather-related time series, including precipitation, rainfall duration, specific humidity, relative humidity, temperature, and others. It is clear that all of these variables are not linearly related, neither do they shift in the same direction, even if they are strongly correlated. For example, we consider the $15^{th}$ column, which encodes an increase in the *precipitation amount*. According to the adjacency matrix, this feature will possibly lead to future decreases in the all day solar radiation, temperature, while causing future increases in specific and relative humidities, etc. From a purely climate-scientific point-of-view, all these points could be accurately verified. On the other hand, the learned adjacency matrix for attention tends to ignore the information brought by precipitation, leading to a loss of accuracy.

Table 1: Multivariate time series forecasting results with prediction length $S \in \{48, 96, 192, 336\}$ for three indices and $S \in \{96, 192, 336, 720\}$ for others. The look back window length is fixed to $T = 96$. The best result are highlighted in red and the second best are in blue and underlined

| Models | | SPACE (Ours) | | iTransformer 2024 | | RLinear 2023 | | PatchTST 2023 | | Crossformer 2023 | | TiDE 2023 | | TimesNet 2023 | | Dlinear 2023 | | Stationary 2022b | |
|---|---|---|---|---|---|---|---|---|---|---|---|---|---|---|---|---|---|---|---|
| Metrics | | MSE | MAE | MSE | MAE | MSE | MAE | MSE | MAE | MSE | MAE | MSE | MAE | MSE | MAE | MSE | MAE | MSE | MAE |
| ETTm1 | 96 | 0.317 | 0.352 | 0.334 | 0.368 | 0.355 | 0.376 | 0.329 | 0.367 | 0.404 | 0.426 | 0.364 | 0.387 | 0.338 | 0.375 | 0.345 | 0.372 | 0.386 | 0.398 |
| | 192 | 0.364 | 0.376 | 0.377 | 0.391 | 0.391 | 0.392 | 0.367 | 0.385 | 0.450 | 0.451 | 0.398 | 0.404 | 0.374 | 0.387 | 0.380 | 0.389 | 0.459 | 0.444 |
| | 336 | 0.395 | 0.397 | 0.426 | 0.420 | 0.424 | 0.415 | 0.399 | 0.410 | 0.532 | 0.515 | 0.428 | 0.425 | 0.410 | 0.411 | 0.413 | 0.413 | 0.495 | 0.464 |
| | 720 | 0.455 | 0.433 | 0.491 | 0.459 | 0.487 | 0.450 | 0.454 | 0.439 | 0.666 | 0.589 | 0.487 | 0.461 | 0.478 | 0.450 | 0.474 | 0.453 | 0.585 | 0.516 |
| | Avg | 0.382 | 0.390 | 0.407 | 0.410 | 0.414 | 0.407 | 0.387 | 0.400 | 0.513 | 0.496 | 0.419 | 0.419 | 0.400 | 0.406 | 0.403 | 0.407 | 0.481 | 0.456 |
| ETTm2 | 96 | 0.170 | 0.251 | 0.180 | 0.264 | 0.182 | 0.265 | 0.175 | 0.259 | 0.287 | 0.366 | 0.207 | 0.305 | 0.187 | 0.267 | 0.193 | 0.292 | 0.192 | 0.274 |
| | 192 | 0.236 | 0.295 | 0.250 | 0.309 | 0.246 | 0.304 | 0.241 | 0.302 | 0.414 | 0.492 | 0.290 | 0.364 | 0.249 | 0.309 | 0.284 | 0.362 | 0.280 | 0.339 |
| | 336 | 0.300 | 0.335 | 0.311 | 0.348 | 0.307 | 0.342 | 0.305 | 0.343 | 0.597 | 0.542 | 0.377 | 0.422 | 0.321 | 0.351 | 0.369 | 0.427 | 0.334 | 0.361 |
| | 720 | 0.402 | 0.395 | 0.412 | 0.407 | 0.407 | 0.398 | 0.402 | 0.400 | 1.730 | 1.042 | 0.558 | 0.524 | 0.408 | 0.403 | 0.554 | 0.522 | 0.417 | 0.413 |
| | Avg | 0.277 | 0.319 | 0.288 | 0.332 | 0.286 | 0.327 | 0.281 | 0.326 | 0.757 | 0.610 | 0.358 | 0.404 | 0.291 | 0.333 | 0.350 | 0.401 | 0.306 | 0.347 |
| ETTh1 | 96 | 0.377 | 0.389 | 0.386 | 0.405 | 0.386 | 0.395 | 0.414 | 0.419 | 0.423 | 0.448 | 0.479 | 0.464 | 0.384 | 0.402 | 0.386 | 0.400 | 0.513 | 0.491 |
| | 192 | 0.426 | 0.418 | 0.441 | 0.436 | 0.437 | 0.424 | 0.460 | 0.445 | 0.525 | 0.492 | 0.525 | 0.492 | 0.436 | 0.429 | 0.437 | 0.432 | 0.534 | 0.504 |
| | 336 | 0.467 | 0.441 | 0.487 | 0.458 | 0.479 | 0.446 | 0.501 | 0.466 | 0.570 | 0.546 | 0.565 | 0.515 | 0.491 | 0.469 | 0.481 | 0.459 | 0.588 | 0.535 |
| | 720 | 0.464 | 0.462 | 0.503 | 0.491 | 0.481 | 0.470 | 0.500 | 0.488 | 0.653 | 0.621 | 0.594 | 0.558 | 0.521 | 0.500 | 0.519 | 0.516 | 0.643 | 0.616 |
| | Avg | 0.433 | 0.427 | 0.454 | 0.447 | 0.446 | 0.434 | 0.469 | 0.454 | 0.529 | 0.522 | 0.541 | 0.507 | 0.458 | 0.450 | 0.456 | 0.452 | 0.570 | 0.537 |
| ETTh2 | 96 | 0.281 | 0.330 | 0.297 | 0.349 | 0.288 | 0.338 | 0.302 | 0.348 | 0.745 | 0.584 | 0.400 | 0.440 | 0.340 | 0.374 | 0.333 | 0.387 | 0.476 | 0.458 |
| | 192 | 0.371 | 0.388 | 0.380 | 0.400 | 0.374 | 0.390 | 0.388 | 0.400 | 0.877 | 0.656 | 0.528 | 0.509 | 0.402 | 0.414 | 0.477 | 0.476 | 0.512 | 0.493 |
| | 336 | 0.410 | 0.426 | 0.428 | 0.432 | 0.415 | 0.426 | 0.426 | 0.433 | 1.043 | 0.731 | 0.643 | 0.571 | 0.452 | 0.452 | 0.594 | 0.541 | 0.552 | 0.551 |
| | 720 | 0.417 | 0.442 | 0.427 | 0.445 | 0.420 | 0.440 | 0.431 | 0.446 | 1.104 | 0.763 | 0.874 | 0.679 | 0.462 | 0.468 | 0.831 | 0.657 | 0.562 | 0.560 |
| | Avg | 0.370 | 0.396 | 0.383 | 0.407 | 0.374 | 0.398 | 0.387 | 0.407 | 0.942 | 0.684 | 0.611 | 0.550 | 0.414 | 0.427 | 0.559 | 0.515 | 0.526 | 0.516 |
| Exchange | 96 | 0.084 | 0.204 | 0.086 | 0.206 | 0.093 | 0.217 | 0.088 | 0.205 | 0.256 | 0.367 | 0.094 | 0.218 | 0.107 | 0.234 | 0.088 | 0.218 | 0.111 | 0.237 |
| | 192 | 0.176 | 0.299 | 0.177 | 0.299 | 0.184 | 0.307 | 0.176 | 0.299 | 0.470 | 0.509 | 0.184 | 0.307 | 0.226 | 0.344 | 0.176 | 0.315 | 0.219 | 0.335 |
| | 336 | 0.324 | 0.413 | 0.331 | 0.417 | 0.351 | 0.432 | 0.301 | 0.397 | 1.268 | 0.883 | 0.349 | 0.431 | 0.367 | 0.448 | 0.313 | 0.427 | 0.476 | 0.476 |
| | 720 | 0.804 | 0.678 | 0.847 | 0.691 | 0.886 | 0.714 | 0.901 | 0.714 | 1.767 | 1.068 | 0.852 | 0.698 | 0.964 | 0.746 | 0.839 | 0.695 | 1.092 | 0.769 |
| | Avg | 0.347 | 0.398 | 0.360 | 0.403 | 0.378 | 0.417 | 0.367 | 0.404 | 0.940 | 0.707 | 0.370 | 0.413 | 0.416 | 0.443 | 0.354 | 0.414 | 0.461 | 0.454 |
| Weather | 96 | 0.165 | 0.205 | 0.174 | 0.214 | 0.192 | 0.232 | 0.177 | 0.218 | 0.158 | 0.230 | 0.202 | 0.261 | 0.172 | 0.220 | 0.196 | 0.255 | 0.173 | 0.338 |
| | 192 | 0.218 | 0.252 | 0.221 | 0.254 | 0.240 | 0.271 | 0.225 | 0.259 | 0.206 | 0.277 | 0.242 | 0.298 | 0.219 | 0.261 | 0.237 | 0.296 | 0.245 | 0.340 |
| | 336 | 0.272 | 0.291 | 0.278 | 0.296 | 0.292 | 0.307 | 0.278 | 0.297 | 0.272 | 0.335 | 0.287 | 0.335 | 0.280 | 0.306 | 0.283 | 0.335 | 0.321 | 0.328 |
| | 720 | 0.347 | 0.340 | 0.358 | 0.347 | 0.364 | 0.353 | 0.354 | 0.348 | 0.398 | 0.418 | 0.351 | 0.386 | 0.365 | 0.359 | 0.345 | 0.381 | 0.414 | 0.355 |
| | Avg | 0.251 | 0.272 | 0.258 | 0.278 | 0.272 | 0.291 | 0.259 | 0.281 | 0.259 | 0.315 | 0.271 | 0.320 | 0.259 | 0.287 | 0.265 | 0.317 | 0.288 | 0.340 |
| Index-1 | 96 | 0.196 | 0.266 | 0.205 | 0.259 | 0.237 | 0.300 | 0.180 | 0.244 | 0.436 | 0.419 | 0.261 | 0.310 | 0.249 | 0.300 | 0.232 | 0.304 | 0.254 | 0.249 |
| | 192 | 0.235 | 0.286 | 0.260 | 0.301 | 0.311 | 0.362 | 0.277 | 0.303 | 0.627 | 0.503 | 0.287 | 0.333 | 0.279 | 0.339 | 0.293 | 0.363 | 0.321 | 0.312 |
| | 336 | 0.300 | 0.347 | 0.367 | 0.378 | 0.541 | 0.532 | 0.310 | 0.356 | 0.644 | 0.553 | 0.323 | 0.368 | 0.354 | 0.407 | 0.512 | 0.525 | 0.527 | 0.422 |
| | 720 | 0.372 | 0.411 | 0.416 | 0.434 | 0.855 | 0.706 | 0.358 | 0.413 | 0.650 | 0.599 | 0.396 | 0.443 | 0.447 | 0.485 | 0.751 | 0.662 | 0.570 | 0.442 |
| | Avg | 0.276 | 0.327 | 0.312 | 0.343 | 0.486 | 0.475 | 0.281 | 0.329 | 0.589 | 0.518 | 0.316 | 0.364 | 0.332 | 0.383 | 0.447 | 0.463 | 0.418 | 0.381 |
| Index-2 | 96 | 0.262 | 0.308 | 0.265 | 0.290 | 0.273 | 0.313 | 0.255 | 0.292 | 0.315 | 0.412 | 0.314 | 0.340 | 0.285 | 0.315 | 0.384 | 0.397 | 0.309 | 0.307 |
| | 192 | 0.313 | 0.354 | 0.290 | 0.324 | 0.333 | 0.368 | 0.294 | 0.325 | 0.355 | 0.423 | 0.351 | 0.375 | 0.315 | 0.372 | 0.461 | 0.452 | 0.410 | 0.357 |
| | 336 | 0.346 | 0.394 | 0.371 | 0.394 | 0.513 | 0.514 | 0.358 | 0.396 | 0.455 | 0.408 | 0.403 | 0.429 | 0.399 | 0.419 | 0.703 | 0.588 | 0.622 | 0.489 |
| | 720 | 0.453 | 0.474 | 0.513 | 0.479 | 0.655 | 0.586 | 0.466 | 0.476 | 0.602 | 0.484 | 0.483 | 0.491 | 0.502 | 0.503 | 0.960 | 0.718 | 0.822 | 0.531 |
| | Avg | 0.343 | 0.382 | 0.360 | 0.372 | 0.444 | 0.445 | 0.343 | 0.372 | 0.432 | 0.432 | 0.388 | 0.409 | 0.375 | 0.402 | 0.627 | 0.539 | 0.541 | 0.421 |
| Index-3 | 96 | 0.364 | 0.376 | 0.424 | 0.377 | 0.367 | 0.380 | 0.490 | 0.385 | 0.976 | 0.720 | 0.456 | 0.414 | 0.456 | 0.414 | 0.380 | 0.394 | 0.463 | 0.402 |
| | 192 | 0.448 | 0.426 | 0.487 | 0.419 | 0.489 | 0.469 | 0.496 | 0.409 | 1.309 | 0.885 | 0.483 | 0.447 | 0.555 | 0.474 | 0.455 | 0.447 | 0.778 | 0.502 |
| | 336 | 0.637 | 0.539 | 0.558 | 0.481 | 0.711 | 0.583 | 0.742 | 0.520 | 1.500 | 0.938 | 0.639 | 0.541 | 0.616 | 0.535 | 0.697 | 0.584 | 1.091 | 0.521 |
| | 720 | 0.752 | 0.631 | 0.759 | 0.619 | 0.978 | 0.713 | 0.762 | 0.629 | 1.660 | 0.987 | 0.844 | 0.690 | 1.136 | 0.771 | 1.020 | 0.739 | 1.107 | 0.589 |
| | Avg | 0.550 | 0.493 | 0.557 | 0.474 | 0.636 | 0.536 | 0.623 | 0.486 | 1.361 | 0.883 | 0.597 | 0.521 | 0.691 | 0.548 | 0.638 | 0.541 | 0.860 | 0.504 |
| $1^{st}Count$ | | 35 | 34 | 2 | 4 | 0 | 1 | 5 | 5 | 2 | 0 | 0 | 0 | 0 | 0 | 1 | 0 | 0 | 1 |

Table 2: Ablation of different components in our model

| Dataset | | ETT | | | | Weather | | | |
|---|---|---|---|---|---|---|---|---|---|
| Prediction Length | | 96 | 192 | 336 | 720 | 96 | 192 | 336 | 720 |
| SPACE | MSE | 0.286 | 0.349 | 0.393 | 0.434 | 0.165 | 0.218 | 0.272 | 0.347 |
| | MAE | 0.330 | 0.369 | 0.400 | 0.433 | 0.205 | 0.252 | 0.291 | 0.340 |
| Attn Instead of TE | MSE | 0.298 | 0.359 | 0.402 | 0.440 | 0.173 | 0.221 | 0.276 | 0.353 |
| | MAE | 0.341 | 0.378 | 0.408 | 0.439 | 0.212 | 0.256 | 0.296 | 0.348 |
| W/O-Seq-Enhancer | MSE | 0.292 | 0.353 | 0.397 | 0.438 | 0.179 | 0.217 | 0.273 | 0.351 |
| | MAE | 0.336 | 0.373 | 0.405 | 0.435 | 0.210 | 0.245 | 0.288 | 0.340 |
| W/O-Encoder | MSE | 0.309 | 0.370 | 0.414 | 0.456 | 0.198 | 0.244 | 0.298 | 0.373 |
| | MAE | 0.351 | 0.389 | 0.418 | 0.451 | 0.233 | 0.273 | 0.312 | 0.362 |

In fact, the attention adjacency matrix is clearly unable to definitively learn features that should contribute to variances in weather patterns. In contrast, the proposed model using TE to model cross-series dependencies can better cope with this situation.

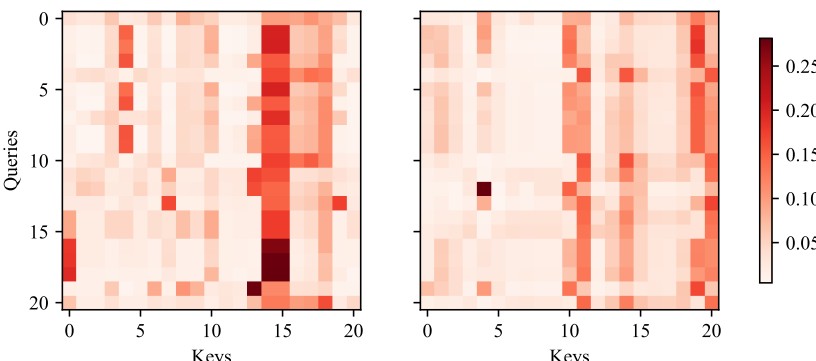

Figure 3: Left: Learned adjacency matrix by Cross TE. Right: Learned matrix by conventional attention mechanism.

### 4.3 ABLATION STUDY

#### 4.3.1 STUDY ON DESIGNED COMPONENTS

We conduct ablation study on datasets includes 4 **ETT** datasets and **Weather** dataset, for which our model performs relatively well even after removal of corresponding modules. We perform three sets of ablation experiments: **Attn Instead of TE**, where transfer entropy we used to calculate cross series dependencies is replaced by attention mechanism, **W/O-Seq-Enhancer** where the module used to enhance sequence data before calculating TE is removed, and **W/O-Encoder** which was done mainly to find out the degree to which performance is impacted by the removal of the TE module. From the results in Table 2, we observe:

- The sequence enhancer is relatively less important, although it can still cause some fluctuations in the model performance if we do not use it;

- Our causal module, containing Cross-TE and CGN, contributes greatly to the performance or SPACE, as expected. By exchanging the TE module with a conventional attention module, we see large increase in MSE across the two datasets. We argue that this is a clear sign of the importance of causality in improving forecasting performance of SPACE compared to baseline models.

#### 4.3.2 HYPERPARAMETER SENSITIVITY

We evaluate the hyperparameter sensitivity of SPACE with respect to the following factors: the learning rate, hidden dimension of each feed forward network, and number of the encoder blocks on six well received baseline datasets. The results shown in Fig. 4 demonstrates that our model is able to maintain a stable performance when parameters are varied, or in other words, the SOTA performance of SPACE is robust against variances in model hyperparameters.

## 5 CONCLUSION

Conventional time-series forecasting models, such as the recent ones based on the attention mechanism, predominantly learn correlative information between time series data. On the other hand, it has been shown in previous work that the analysis of time series based on the learning of causative factors yields better results than models based on correlative ones. Drawing on this fact, we design and implement *SPACE*, a time-series analysis model which learns *causative* information and uses this for downstream forecasting tasks. We introduce several novel modules which significantly simplify the computation, as well as organizes and aggregates this information. We perform extensive experiments which validates our approach. Our experimentations show that:

- SPACE is able to outperform a number of SOTA baseline models in for both the MSE and MAE metrics, and hence is shown to be superior for general forecasting tasks;

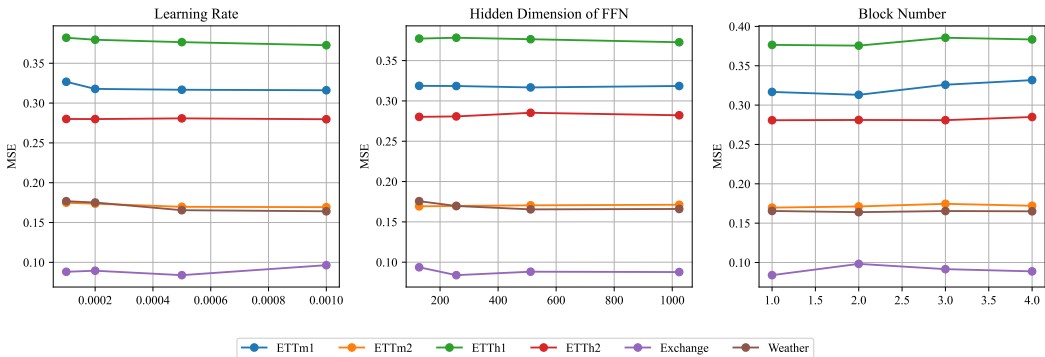

Figure 4: Hyperparameter sensitivity with respect to the learning rate, hidden dimension of each feed forward network, and number of the encoder blocks. All the results are computed with the look back window length set to $S = 96$, and predict window length $P = 96$

- The computed adjacency matrices show that learned features for the **Weather** dataset show good correspondence with known cause-and-effects from climate modelling and forecasting;

- Ablation studies clearly show the importance of the Cross-TE and CGNN modules in helping our framework achieve SOTA performance in multivariate time-series forecasting.

In short, we believe that SPACE paves the way for the design of more robust and consistent forecasting models based on causative information.

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

## A  IMPLEMENTATION DETAILS

Our experiments on all conducted on a single NVIDIA RTX 4090 24GB GPU, and code is implemented in PyTorch which can be found in https://anonymous.4open.science/r/EntroNet-4B05.

### A.1  DATASET DESCRIPTION

We conduct our experiment on 9 real-world datasets, including (1) **ETT** Zhou et al. (2021) contains 4 datasets with 7 sub-series of electricity data from July 2016 to July 2018 in it. ETTh1 and ETTh2 are two of the datasets recorded every hour, while ETTm1 and ETTm2 are two recorded every 15 minutes. (2)**Weather** Wu et al. (2021) contains 21 indicators of weather condition, such as air temperature and humidity, which are recorded every 10 minutes in 2020 from the Weather Station of the Max Planck Biogeochemistry Institute. (3)**Exchange-rate** Wu et al. (2021) is a dataset collecting daily exchange rates from 8 countries from 1990 to 2016.

In addition to the commonly used public datasets that serve as benchmarks for time series prediction methodologies, we have also curated three proprietary financial indices datasets to further supplement our evaluation. The **Financial Indices** dataset comprises three distinct financial indices sourced from SSEC, SZI, and CSI500, providing additional depth and relevance to our analysis.

As for the forecasting settings, we fix the look-back window length to 96 across all datasets. The prediction lengths are set at 48, 96, 192, 336 for the three Financial Indices, and 96, 192, 336, 720 for the remaining datasets. Consistent with prior methodologies such as iTransformer Liu et al. (2023), we adhere to the same data processing procedures and maintain the same train-validation-test split order. In terms of data leakage issue, we have rigorously maintained the chronological order of the training, validation, and test datasets, ensuring that no future information will be leaked to training process.

### A.2  IMPLEMENTATION DETAILS

---

**Algorithm 1** EntroNet-Overall Architecture

---

**Require:** Input series $\mathbf{X} \in R^{S \times N}$; Input series length S; Number of variates N; Prediction length L; Patch number or time states number T; Patch length PL; Number of encoder layers EL; Number of graph layers GL.

1: Initialize the variables
2: $\mathbf{X} = \mathbf{X}^{\top}$ $\{\mathbf{X} \in \mathbb{R}^{N \times S}\}$
3: ▷ Unfold the series in the last dimension in order to generate patches.
4: $\mathbf{X} = \text{UnFold}(\mathbf{X})$ $\{\mathbf{X} \in \mathbb{R}^{N \times T \times PL}\}$
5: ▷ Project $\mathbf{X}$ into embedding $\mathbf{H_0}$ on the last dimension.
6: $\mathbf{H_0} = \mathbf{X} \cdot \mathbf{W} + \mathbf{b}$ $\{\mathbf{H_0} \in \mathbb{R}^{N \times T \times d}\}$
7: ▷ Sequence Enhancer using mere multihead attention to enhance series.
8: $\mathbf{H} = \mathbf{H_0} + \text{Multihead-Attention}(\mathbf{H_0})$ $\{\mathbf{H} \in \mathbb{R}^{N \times T \times d}\}$
9: **for** i in $\{1, 2, ..., \text{EL}\}$ **do**
10:    ▷ **Cross TE** module calculating pseudo transfer entropy matrix among series. Output $\mathbf{T}$ can be denoted as $\{t_{ij}\}_{N \times N}$ where $t_{ij}$ denotes causality from series $j$ to $i$.
11:    $\mathbf{T} = \text{Fast-pTE}(\mathbf{H})$ $\{\mathbf{T} \in \mathbb{R}^{N \times N}\}$
12:    ▷ Causality-based Graph Network **CGN** which aggregates information.
13:    **for** $l$ in $\{1, 2, ..., \text{GN}\}$ **do**
14:       $\mathbf{H}^{(1)} = \text{2dMixer}(H^{(l-1)})$
15:    **end for**
16:    $\mathbf{H} = \mathbf{H} + \mathbf{T} \cdot \mathbf{H}^{(GN)}$ $\{\mathbf{H} \in \mathbb{R}^{N \times T \times d}\}$
17: **end for**
18: ▷ **Projector**
19: $\mathbf{O} = \text{Flatten}(\mathbf{H}) \cdot \mathbf{W} + \mathbf{b}$ $\{\mathbf{O} \in \mathbb{R}^{N \times L}\}$
20: If no errors return =0

---

# B  FAST pTE

## B.1  ALGORITHM

Pseudo transfer entropy (pTE) is a rigorous algorithm that requires precise alignment of each observation in the two sequences according to their temporal order, i.e., given two time series $\mathbf{x} = \{x_1, x_2, ..., x_T\}$ and $\mathbf{y} = \{y_1, y_2, ..., y_T\}$, each $x_i$ and $y_i$ should occur at the same time point, while real-world data cannot be so precise.

Moreover, pTE can sometimes make mistakes due to the complex dynamic property of time series, which can be limited by the sampling of series data, and fails to capture information flows on multiple time scales.

Therefore, we choose to calculate TE on time steps, i.e., on the patches instead of the initial time points. This method have two advantages: (1) It enables the model to learn causality in a hidden state with higher dimension, allowing its flexibility and stronger capability. (2) It is more robust and less easy to be influenced by the precision of measurement than pure TE computation method. (3) With linear projection on each patch in previous steps, it can make use of the multi-scale dynamics of time series. We shall prove the third one in next section.

Finally, although pTE has fairly reduce the computational cost of original TE, it can still be a bottleneck for our training process. To state it more clearly, we consider the random process $\mathbf{X}$ follows the normal distribution. Therefore, the entropy of a p-variate normal variable, e.g. $\mathbf{x} \sim \mathcal{N}(\mathbf{x}|\mu, \mathbf{\Sigma})$, is given by:

$$
\begin{aligned}
H_d(\mathbf{x}) &= - \int_{R^p} \mathcal{N}(\mathbf{x}|\mu, \mathbf{\Sigma}) \log \mathcal{N}(\mathbf{x}|\mu, \mathbf{\Sigma}) \, \mathrm{d}\mathbf{x} \\
&= \frac{1}{2}[p + p \log 2\pi + \log(|\mathbf{\Sigma}|)]
\end{aligned}
\tag{7}
$$

And pTE can be re-written as:

$$
H(i_n^{(k)}, j_n^{(l)}) - H(i_{n+1}, i_n^{(k)}, j_n^{(l)}) + H(i_{n+1}, i_n^{(k)}) - H(i_n^{(k)})
\tag{8}
$$

With the two equations provided, it is natural to derive the formula for pTE stated before. Full detail can be found in Silini & Masoller (2021a).

We can conclude from the above equations that the vital part in calculating traditional pTE is to get the covariance of $\mathbf{I}_t \oplus \mathbf{J}_t$, $\mathbf{i} \oplus \mathbf{I}_t$, $\mathbf{i} \oplus \mathbf{I}_t \oplus \mathbf{J}_t$ and $\mathbf{I}_t$ as stated in Preliminaries of Methodology. Original algorithm for pTE are provided as the following Algorithm 2.

**Embed** function are shown as below:

---

**Algorithm 3** Embed Function

---

0:  $(ch, N) \leftarrow \text{shape}(x)$
0:  $hidx \leftarrow \text{arange}(0, \text{nt} \times \text{lag}, \text{step} = \text{lag})$
0:  $Nv \leftarrow N - (\text{nt} - 1) \times \text{lag}$
0:  $u \leftarrow \text{zeros}(\text{nt} \times ch, Nv)$
0:  **for** $i = 0$ **to** $\text{nt} - 1$ **do**
0:      $u[i \times ch : (i+1) \times ch] \leftarrow x[:, hidx[i] : hidx[i] + Nv]$
0:  **end for**
0:  If no errors return =0

---

Full detail can be found in our code. In the algorithm, the time complexity to compute $\Sigma$ over two series is $O(d^2 T)$, with the multiplication of two matrices of shape $[3d \times (T - nt \cdot lag)]$. It will be really time consuming if we set the $d$ relatively large and can be easily out of memory. Therefore, we propose our Fast-pTE algorithm in order to lower its cost. The general process of computation is almost the same with original pTE, however, with one flatten step before calculating $\Sigma$. The input series will be first flatten to a 2d matrix, and then forward to the following steps. The complexity will be reduced to $O(dT)$, with matrix multiplication of shape $[3 \times Td]$.

---

**Algorithm 2** pTE

---

**Require:** Input enhanced series $\mathbf{Q}$ and $\mathbf{K} \in R^{N \times d \times T}$; Input variate number N; Input hidden dimension d; Number of time states T; Number of $\mathbf{I}_t$ nt; Time delay lag.
1: Initialize the variables
2: Remove-trend($\mathbf{H}$) $\{\mathbf{H} \in \mathbb{R}^{N \times d \times T}\}$
3: ▷ Generate $\mathbf{i}, \mathbf{I_t}, \mathbf{J_t}$ with function **Embed**
4: $\mathbf{Q}_{embed} = \text{Embed}(\mathbf{Q})$ $\{\mathbf{Q}_{embed} \in \mathbb{R}^{N \times (d \times (1+nt)) \times (T-nt \cdot lag)}\}$
5: $\mathbf{K}_{embed} = \text{Embed}(\mathbf{K})[:, :, : -1]$ $\{\mathbf{K}_{embed} \in \mathbb{R}^{N \times (d \cdot nt) \times (T-nt \cdot lag)}\}$
6: ▷ Concatenate along dimension one and compute covariance for all series as a whole.
7: $\mathbf{H} = \text{Concat}([\mathbf{Q}, \mathbf{K}])$
8: $\mathbf{avg} = \text{Mean}(\mathbf{H}, \text{dim=1})$
9: $\Sigma = (\mathbf{H} \cdot \mathbf{H}.permute(0, 2, 1))/\mathbf{avg}$
10: ▷ Select $\Sigma(\mathbf{I}_t \oplus \mathbf{J}_t), \Sigma(\mathbf{i} \oplus \mathbf{I}_t), \Sigma(\mathbf{i} \oplus \mathbf{I}_t \oplus \mathbf{J}_t)$ and $\Sigma(\mathbf{I}_t)$ from $\Sigma$, with shape $[N \times N \times 2nt \times 2nt], [N \times (1 + nt) \times 1 + nt], [N \times N \times (2nt + 1) \times (2nt + 1)], [N \times 1]$ subsequently.
11: ▷ $\mathbf{H}_i(i = 1, 2, 3, 4)$ are determinant of the above four covariance matrix.
12: **for** $i, \Sigma'$ in enumerate($\{\Sigma(\mathbf{I}_t \oplus \mathbf{J}_t), \Sigma(\mathbf{i} \oplus \mathbf{I}_t), \Sigma(\mathbf{i} \oplus \mathbf{I}_t \oplus \mathbf{J}_t)$ and $\Sigma(\mathbf{I}_t)\}$) **do**
13: $\quad \Sigma' = \text{Select-from}(\Sigma)$
14: $\quad \mathbf{H}_i = det(\Sigma')$
15: $\quad ▷ \mathbf{H}_i \in \mathbb{R}^{N \times N} if i = 1, 3$
16: $\quad ▷ \mathbf{H}_i \in \mathbb{R}^{N \times 1} if i = 2, 4$
17: **end for**
18: Calculate pTE $\{pTE \in \mathbb{R}^{N \times N}\}$
19: If no erros return =0

---

**Algorithm 4** Fast-pTE

---

**Require:** Input series $\mathbf{Q}$ and $\mathbf{K} \in \mathbb{R}^{N \times d \times T}$
1: $\mathbf{Q} = \text{Flatten}(\mathbf{Q}.permute(0, 2, 1))$ $\{\mathbf{Q} \in \mathbb{R}^{N \times T \cdot d}\}$
2: $\mathbf{K} = \text{Flatten}(\mathbf{K}.permute(0, 2, 1))$ $\{\mathbf{K} \in \mathbb{R}^{N \times T \cdot d}\}$
3: fast-pTE = pTE($\mathbf{Q}, \mathbf{K}$) $\{\text{fast-pTE} \in \mathbb{R}^{N \times N}\}$
4: If no errors return =0

---

Next, we would like to prove that the Fast-pTE is the same as pTE.

### B.2 MULTI-SCALE NATURE OF CROSS TE METHOD

Previous works correspond to multi-scale transfer entropy mainly use moving average as their way to take more data points, i.e., different scales, into account. However, this approach has limited the capability of detecting multi-scale dynamics since if we view it from the perspective of interpolation, it merely considers the middle point of a range of time points. It can be inferred that a linear mapping is a stronger way to capture the multi-scale dynamics, as we extrapolate the average to linear interpolation.

Consider linear map $\mathbf{W} = [w_1, w_2, w_d]\top$, and the patches before embedding is $\mathbf{P} = [p_1, p_2, ..., p_T]$, where $w_i \in \mathbb{R}^{1 \times n}$ and $p_j \in \mathbb{R}^{n \times 1}$, which will then be embedded as $\tilde{p}_{ij} = w_i \cdot p_j$. Hence if we normalise the $w_i$ to $\tilde{w}_i = \frac{w_i}{|w_i|}$, we can re-write embedding function $\tilde{p}_{ij} = |w_i|\tilde{w}_i \cdot p_j$, where $\tilde{w}_i \cdot p_j$ is a linear interpolation. In addition, $\forall i, j \in \{1, 2, ..., T\}, p_{ki}$ and $p_{kj}$ are mapped with the same $w_j$, preserving the time order for $\tilde{p}_{ki}$ and $\tilde{p_{kj}}$. Therefore, with the linear interpolation that enlarge the horizons depends on different value of $w_i$ while perpetuate the temporal order, this method can theoretically acceptable in calculating multi-scale transfer entropy.

Back to our model, except for embedding, there are several other steps that have been implemented, which could potentially undermine the aforementioned advantages. However, by retaining the residual connections, we ensure that the original information is preserved throughout the calculations, thereby mitigating this issue. To be more specific, there are two non-linear mapping in the steps that might be happen earlier than a Cross TE module, such as Feed-Forward layer and 2dMixer. Admittedly, it will break the linear interpolation which we discussed above, but in each step we use the

non-linear function, we apply residual connection $\tilde{\mathbf{H}} = \mathbf{H} + \text{non-linear}(\mathbf{H})$. Hence it is observed that the time information within and across patches can all be remembered by the model, which indicates the practicality of the method.

### B.3 Fast-pTE Outperforms Original pTE

Though Fast-pTE can reduce the computation cost by a large amount, we will show that they have done the same task in the context of our discussion though our fast one saves computational cost by a large amount. For simplicity, we ignore the bias part in the discussion.

First, based on the discussion of section **Multi-scale Nature of Cross TE Method**, we know that each point in the inputted data is a interpolation of original one. Hence if we flatten the 2d series, all data point can be seen as a value of a new time point, and each variables are interpolated in the same way. Therefore, the flattened series is an expansion of the original one. With no information exchange between future values and past values, calculating TE on this series is an appropriate approach.

Next, we want to show the resemblance between them using matrix operations. Consider two time series $\mathbf{q}$ and $\mathbf{k} \in \mathbb{R}^{d \times T}$, where $d$ is the dimension of series and $T$ is the number of time steps, which are linear projections of $\mathbf{y}$ and $\mathbf{x} \in \mathbb{R}^{d \times T}$ with linear map $\mathbf{W}_q$ and $\mathbf{W}_k \in \mathbb{R}^{d \times d}$.

Rigorously, to calculate pTE, we need to find past value of $\mathbf{q}$ named $\mathbf{q}_t$, past value of $\mathbf{k}$ named $\mathbf{k}_t$, and future value of $q$, denoted as $\mathbf{q}_f$. According to our description in Algorithm part, we know that the above three matrices is a kind of embedding that simply changes the position of elements in its original one. Hence, we observe that

$$
\begin{aligned}
\mathbf{q}_t &= \mathbf{W}_q \cdot \mathbf{y}[:, :-1] = \mathbf{W}_q \cdot \mathbf{y}_t \\
\mathbf{k}_t &= \mathbf{W}_k \cdot \mathbf{x}[:, :-1] = \mathbf{W}_k \cdot \mathbf{x}_t \\
\mathbf{q}_f &= \mathbf{W}_q \cdot \mathbf{y}[:, 1:] = \mathbf{W}_q \cdot \mathbf{y}_f
\end{aligned}
\tag{9}
$$

Based on the equations above, we can derive the formula for $\mathbf{I}_t \oplus \mathbf{J}_t$, $\mathbf{i} \oplus \mathbf{I}_t$, $\mathbf{i} \oplus \mathbf{I}_t \oplus \mathbf{J}_t$ and $\mathbf{I}_t$.

$$
\begin{aligned}
\mathbf{I}_t \oplus \mathbf{J}_t &= \begin{bmatrix} \mathbf{W}_q & \mathbf{0} \\ \mathbf{0} & \mathbf{W}_k \end{bmatrix} \cdot \begin{bmatrix} \mathbf{y}_t \\ \mathbf{x}_t \end{bmatrix} \\
&= \mathbf{W}_1 \cdot \begin{bmatrix} \mathbf{y}_t \\ \mathbf{x}_t \end{bmatrix}
\end{aligned}
\tag{10}
$$

Similarly, the remaining matrices can also be re-written as

$$
\begin{aligned}
\mathbf{i} \oplus \mathbf{I}_t &= \mathbf{W}_2 \cdot \begin{bmatrix} \mathbf{y}_f \\ \mathbf{y}_t \end{bmatrix} \\
\mathbf{i} \oplus \mathbf{I}_t \oplus \mathbf{J}_t &= \mathbf{W}_3 \cdot \begin{bmatrix} \mathbf{y}_f \\ \mathbf{y}_t \\ \mathbf{x}_t \end{bmatrix} \\
\mathbf{I}_t &= \mathbf{W}_4 \cdot \mathbf{yt}
\end{aligned}
\tag{11}
$$

where $\mathbf{W}_1, \mathbf{W}_2, \mathbf{W}_3, \mathbf{W}_4$ are all square matrices.

To calculate pTE, we only need to know the covariance of $\mathbf{I}_t \oplus \mathbf{J}_t$, $\mathbf{i} \oplus \mathbf{I}_t$, $\mathbf{i} \oplus \mathbf{I}_t \oplus \mathbf{J}_t$ and $\mathbf{I}_t$. Note that they are all the linear map of a matrix, hence we can calculate their covariance based on Lemma 1.

#### B.3.1 Lemma 1.

$\forall \mathbf{W} \in \mathbb{R}^{d \times d}$ and $\mathbf{X} \in \mathbb{R}^{d \times T}$, *each row of* $\mathbf{X}$ *represents a different variable, with each column corresponding to an observation of these variables. We find* $Var(\mathbf{WX}) = \mathbf{W} \cdot Var(\mathbf{X}) \cdot \mathbf{W}^\top$ *and* $|Var(\mathbf{WX})| = |Var(\mathbf{X})| \cdot |Var(\mathbf{W})|$.

However, in the Fast-pTE algorithm, the input matrices are first flattened in the way stated in Algorithm 4. It is easy that in this situation

### B.3.2 LEMMA 2.

$\forall \mathbf{W} \in \mathbb{R}^{d \times d}$ and $\forall \mathbf{X} \in \mathbb{R}^{d \times T}$, if we flatten $\mathbf{WX}$ to a 1d vector $\mathbf{y}$, then $Var(\mathbf{y}) = tr(\mathbf{XX}^\top \cdot \mathbf{WW}^\top)$

With Lemma 1 and Lemma 2 (proof will be given in the next section), we compute the determinant of covariance of four matrices $\mathbf{I}_t \oplus \mathbf{J}_t$, $\mathbf{i} \oplus \mathbf{I}_t$, $\mathbf{i} \oplus \mathbf{I}_t \oplus \mathbf{J}_t$ and $\mathbf{I}_t$.

**(1)** *The simplest condition:* $\mathbf{I}_t$.

It can be directly derived from the two lemmas that in the origin pTE

$$|\Sigma(I_t)| = |Var(X)Var(W_4)| \tag{12}$$

While in Fast-pTE

$$|\Sigma(I_t)| = tr(XX^\top W_4 W_4^\top) = tr(\Sigma(I_t)) \tag{13}$$

**(2)** *Complex situations:* $\mathbf{I}_t \oplus \mathbf{J}_t$, $\mathbf{i} \oplus \mathbf{I}_t$, $\mathbf{i} \oplus \mathbf{I}_t \oplus \mathbf{J}_t$

In these situation, covariance of them are correspond with two or more different series, the analyzing processes are the same, so we may wish to simply discuss the $\mathbf{I}_t \oplus \mathbf{J}_t$.

First, denote $\begin{bmatrix} W_q \\ W_k \end{bmatrix} = \begin{bmatrix} w_1 \\ w_2 \\ \vdots \\ w_d \\ w_{d+1} \\ \vdots \\ w_{2d} \end{bmatrix}$ where $w_i \in \mathbb{R}^{1 \times d}$. Hence $\mathbf{I}_t \oplus \mathbf{J}_t = \begin{bmatrix} w_1 y_t \\ w_2 y_t \\ \vdots \\ w_d y_t \\ w_{d+1} x_t \\ \vdots \\ w_{2d} x_t \end{bmatrix}$. Therefore,

$$\Sigma(\mathbf{I}_t \oplus \mathbf{J}_t) = \begin{bmatrix} w_1 yy^\top w_1 & w_1 yy^\top w_2^\top & ... & w_1 yx^\top w_{2d}^\top \\ \vdots & \vdots & & \vdots \\ w_{d+1} xy^\top w_1^\top & w_{d+1} xy^\top w_2^\top & ... & w_{d+1} xx^\top w_{2d}^\top \\ \vdots & \vdots & & \vdots \\ w_{2d} xy^\top w_2^\top & w_{2d} xy^\top w_2^\top & ... & w_{2d} xx^\top w_{2d}^\top \end{bmatrix}$$
$$= \begin{bmatrix} W_q yy^\top W_q^\top & W_q yx^\top W_k^\top \\ W_k xy^\top W_q^\top & W_k xx^\top W_k^\top \end{bmatrix} \tag{14}$$

For Fast-pTE, $\Sigma(\mathbf{I}_t \oplus \mathbf{J}_t)$ are flattened before calculating covariance, which equals to $\begin{bmatrix} w_1 y_t & w_2 y_t & \cdots & w_d y_t \\ w_{d+1} x_t & w_{d+2} x_t & \cdots & w_{2d} x_t \end{bmatrix}$ So

$$\Sigma^{'}(\mathbf{I}_t \oplus \mathbf{J}_t) = \begin{bmatrix} \Sigma_{i=1}^d w_i yy^\top w_i^\top & \Sigma_{i=1}^d w_i yx^\top w_{d+i}^\top \\ \Sigma_{i=1}^d w_{d+i} xy^\top w_d^\top & \Sigma_{i=1}^d w_{d+i} xx^\top w_{d+i}^\top \end{bmatrix}$$
$$= \begin{bmatrix} tr(W_q yy^\top W_q^\top) & tr(W_q yx^\top W_k^\top) \\ tr(W_k xy^\top W_q^\top) & tr(W_k xx^\top W_k^\top) \end{bmatrix} \tag{15}$$

From the discussion, it is observed that the difference between original pTE and our Fast-pTE is almost the same with the difference between determinant and trace of the covariance matrix. The determinant is able to consider all the information in the covariance matrix, including variance of a single variate and covariance between variates, while trace pays attention to only the variance of a single series. Back to our time-series forecasting problems, the initial series inputted into the model are 1d vectors, which are patchfied and embedded to $\mathbb{R}^{d \times T}$. From the perspective of original Fast-pTE algorithm, though it fails to treat different dimension of a series as different variates and find relation among them, it takes them as a single one, only concentrating on relation between two time series outside hidden dimension. In this case, we think Fast-pTE has already complete the main task to model the causality among series, apart from hugely reduce the computational cost.

### B.4  PROOF OF LEMMA

#### B.4.1  PROOF OF LEMMA 1

*Proof.* With the property of covariance of a matrix, it is easy that

$$Var(\mathbf{WX}) = \mathbf{WX} \cdot (\mathbf{WX})^\top = \mathbf{W}Var(\mathbf{X})\mathbf{W}\top \qquad (16)$$

Since $\mathbf{W} \in \mathbb{R}d \times d$ is a square matrix, using the property of determinant of it

$$|Var(\mathbf{WX})| = |Var(\mathbf{X})| \cdot |WW^\top| \qquad (17)$$

$\square$

#### B.4.2  PROOF OF LEMMA 2

*Proof.* Let us consider $\mathbf{W}$ **and** $\mathbf{X}$ in a more detailed way.

We denote $\mathbf{W}$ as $[\mathbf{w_1}^\top, \mathbf{w_2}^\top, ..., \mathbf{w_d}^\top]^\top$ and $\mathbf{X}$ as $[\mathbf{x_1}, \mathbf{x_2}, ..., \mathbf{x_T}]$, hence covariance $\Sigma$ of $\mathbf{WX}$ is given by

$$\begin{bmatrix} w_1 Var(X)w_1^\top & w_1 Var(X)w_2^\top & ... & w_1 Var(X)w_d^\top \\ w_2 Var(X)w_1^\top & w_2 Var(X)w_2^\top & ... & w_2 Var(X)w_d^\top \\ \vdots & \vdots & & \vdots \\ w_d Var(X)w_1^\top & w_d Var(X)w_2^\top & ... & w_d Var(X)_{w_d}^\top \end{bmatrix}$$

After flatten step, the matrix $\mathbf{WX}$ become

$$\mathbf{y} = [w_1 X \quad w_2 X \quad ... \quad w_d X] \qquad (18)$$

Hence, the covariance of $\mathbf{y}$ is given by

$$\begin{aligned} Var(\mathbf{y}) &= \Sigma_{i=1}^d w_i Var(X)w_i^\top \\ &= tr(WXX^\top W^\top) = tr(\Sigma) \\ &= tr(XX^\top \cdot WW^\top) \end{aligned} \qquad (19)$$

$\square$

