# OpenReview forum: "Multivariate Time-series Forecasting with SPACE: Series Prediction Augmented by Causality Estimation"
_ICLR.cc/2025/Conference — Submitted to ICLR 2025_

### Official Review · Reviewer_goHF · 2024-10-29

**Soundness:** 2
**Presentation:** 3
**Contribution:** 3
**Rating:** 5
**Confidence:** 4

**Summary:**

This paper introduces SPACE to forecast MTS enhanced by causality estimation.
Specifically, this method captures causal relationships by a transfer entropy-based cross-TE module and a casual GNN.

**Strengths:**

-	Strong motivation: The paper presents an interesting method by providing a causative perspective for analyzing time series.
-	Good clarity: The writing is clear and well-structured, making the paper easy to follow and understand.

**Weaknesses:**

- Lack detailed comparisons with highly related works.
    - The proposed method relies heavily on patch mixers and time mixers (as described in Equation 6). However, state-of-the-art mixer-based methods, e.g., TimeMixer [1] and Timexer [2], are neither discussed in the related work nor included in the experimental comparisons.
    - Since one of the main contributions is about causal GNN, some GNN related works, e.g., CrossGNN [3], should be mentioned and compared.
- Lack a comprehensive efficiency study.
    - The method introduces the fast-pTE algorithm and emphasize its efficiency advantage in contributions. It is recommended to provide detailed theoretical analysis or discussion and conduct corresponding ablation studies about variants with TE, pTE, and fast-pTE.
    - The efficiency study about the proposed work and SOTA works are also necessary, which can better demonstrate practical benefits of the proposed work.
- Need more evaluations about the learned relationships.
    - The authors claim that Cross TE is “designed to dynamically learn and adapt to evolving causal dependencies while preserving memory of past relationships.” However, the current evaluations are insufficient to convincingly demonstrate this claim. Relying on a single sample of the learned adjacency matrix (as shown in Fig. 3) is not enough to explain the dynamic nature of causality. Additional evaluations focusing on dynamic causal dependencies should be provided.
    - It is also recommended to include examples from real-world datasets that showcase causal relationships. Presenting these examples in a visual format similar to Fig. 1 would help clarify the learned relationships.
- Should enhance presentations.
    - In Fig. 2, subfigures (a) and (b) seem unnecessary. It is more valuable to provide a detailed illustration of the Entropy Graph construction, as it is closely related to the paper’s main contribution.
    - For Fig. 3, adding variable names instead of numbers on the axes can enhance clarity and understanding.

[1] Wang, S., Wu, H., Shi, X., Hu, T., Luo, H., Ma, L., ... & ZHOU, J. 2024. TimeMixer: Decomposable Multiscale Mixing for Time Series Forecasting. In The Twelfth International Conference on Learning Representations.

[2] Wang, Y., Wu, H., Dong, J., Liu, Y., Qiu, Y., Zhang, H., ... & Long, M. 2024. Timexer: Empowering transformers for time series forecasting with exogenous variables. arXiv preprint arXiv:2402.19072.

[3] Huang, Q., Shen, L., Zhang, R., Ding, S., Wang, B., Zhou, Z., & Wang, Y. 2023. CrossGNN: Confronting noisy multivariate time series via cross interaction refinement. Advances in Neural Information Processing Systems, 36, 46885-46902.

**Questions:**

Please see the Weaknesses part.

---

### Official Review · Reviewer_HJwG · 2024-11-02

**Soundness:** 3
**Presentation:** 1
**Contribution:** 2
**Rating:** 3
**Confidence:** 4

**Summary:**

The paper presents the SPACE model, an approach for multivariate time-series forecasting that integrates causality estimation to capture complex interdependencies.

**Strengths:**

1. **Innovative Approach**: Integrates causality estimation with multivariate time-series forecasting, which is a unique perspective.
2. **Real-World Application**: Demonstrates effectiveness in real-world, multivariate forecasting tasks, which may offer practical benefits.

**Weaknesses:**

1. **Dataset Coverage**: Commonly used datasets, such as *Traffic* and *Electricity*, are missing from the evaluation, limiting the scope of comparison.
2. **Efficiency Claims**: While SPACE is described as highly efficient, there is no empirical comparison of runtime with established models like iTransformer and DLinear.
3. **Unverified Claims in Figures**: Figure 1 implies that the attention mechanism only focuses on similar series, a claim not substantiated by experiments or theory.
4. **Notation and Writing Inconsistencies**: The paper has numerous small errors in notation (e.g., non-italicized symbols) and inconsistent symbol usage, particularly in the methodology section (e.g., line 276 “N” should be italic, lines 274 and 284 should use “P_N,” and line 285 should italicize “T”). Such inconsistencies affect readability and technical accuracy.
5. **Causal Adjacency Matrix Evaluation**: It is unclear in Figure 3 how the learned adjacency matrix by Cross-TE outperforms traditional attention in identifying causal relationships. Figure 1 claims attention ignores dissimilar information, yet Figure 3 does not convincingly demonstrate that Cross-TE resolves this.
6. **Clarity in Equation 1**: Equation 1 lacks clarity in summation notation, as it’s unclear which part of the formula is summed over. Additionally, the notation for \(i\) as a time step index raises questions about why an additional superscript is needed to denote the previous time.
7. **Inconsistencies in Variable Definition**: In the problem definition, \(x\) is defined as a time series, \(X\) as historical sequences, and \(Y\) as future sequences. However, line 263 redefines \(y\) as historical, which can be misleading and suggests potential label leakage. The overall presentation of the methodology is unclear and could benefit from consistent use of symbols and fonts.
8. **Perceived Model Complexity**: Without clear experimental support for its causal relationship advantages, SPACE may appear as a combination of existing techniques (PTE, attention, GCN, and Time Mixers) without sufficient innovation in causality extraction.

**Questions:**

1. Can the authors clarify the specific problem in time-series causality extraction that SPACE addresses, beyond combining existing modules?
2. Could additional ablation studies be provided to isolate and validate the causal contributions of Cross-TE versus standard attention mechanisms?
3. Are there any plans to release additional experiments or dataset evaluations, specifically with *Traffic* and *Electricity*, to strengthen model generalizability?

---

### Official Review · Reviewer_MPNx · 2024-11-03

**Soundness:** 2
**Presentation:** 3
**Contribution:** 2
**Rating:** 3
**Confidence:** 4

**Summary:**

The paper presents an end-to-end trainable Series Prediction model Augmented by Causality Estimation, namely SPACE,  to incorporate temporal dependencies and causal relationships in time series forecasting. SPACE utilizes a temporal embedding and a transfer entropy module in the hope to capture the causal structures within multivariate time series for better forecasting.

**Strengths:**

+ Causal structures is an important aspect for time series forecasting. Most exsiting work did not take it into consideration
+ The authors considers scalability when designing the model.
+ The proposed idea is reasonable and some encouraging experiment results.

**Weaknesses:**

- While considering causal structures is a good idea, the way SPACE infers the causal structures is not fully convincing. The paper goes a long way to discuss issues with existing work, especially those utilizes granger causality, ad argues for transfer entropy. Since transfer entropy (TE) is difficult to calculate, the authors proposed to use pseudo TE which assumes that "the time series follow the normal distribution". This is an extremely strong assumption and not "acceptable assumption for real-world time series".

- The improvement SPACE achieves over state-of-art methods are very marginal. It is not clear whether it is worthwhile to go with such a complicated model but very limited improvement.

**Questions:**

The dataset used in the experiments are rather toy small set. What are the experiment results of SPACE on larger dataset, such as NY taxi or climate datasets?

Can you add more recent baselines, such as TSmixture?  https://arxiv.org/abs/2303.06053

---

### Official Review · Reviewer_f9GK · 2024-11-03

**Soundness:** 2
**Presentation:** 3
**Contribution:** 2
**Rating:** 3
**Confidence:** 5

**Summary:**

The paper introduces SPACE (Series Prediction Augmented by Causality Estimation), a novel model for multivariate time series forecasting. SPACE addresses three key characteristics of multivariate time series: causal relationships rather than mere similarities, information across multiple independent factors, and inherent temporal dependencies. The model integrates several components, including a Sequence Enhancer using attention mechanisms, a Cross-TE module that computes transfer entropy to capture causal relationships, and a Causal Graph Neural Network (CGNN) that uses the causality matrix as an adjacency matrix. The authors argue that conventional time series analysis methods often fail to capture these complex relationships, leading to incomplete or misleading conclusions.

**Strengths:**

The paper introduces SPACE (Series Prediction Augmented by Causality Estimation), a novel model for multivariate time series forecasting. SPACE addresses three key characteristics of multivariate time series: causal relationships rather than mere similarities, information across multiple independent factors, and inherent temporal dependencies. The model integrates several components, including a Sequence Enhancer using attention mechanisms, a Cross-TE module that computes transfer entropy to capture causal relationships, and a Causal Graph Neural Network (CGNN) that uses the causality matrix as an adjacency matrix. The authors argue that conventional time series analysis methods often fail to capture these complex relationships, leading to incomplete or misleading conclusions.

Experimental results demonstrate that SPACE outperforms eight state-of-the-art baseline models on nine real-world datasets. The model shows improved performance across various prediction lengths and datasets, including weather, electricity, and financial data. The authors emphasize that the integration of causal information is essential for improving forecasting performance in complex, real-world time series data. Additionally, they claim that SPACE enhances the interpretability of forecasts, especially on weather-related data, by capturing and visualizing causal relationships between different variables.

**Weaknesses:**

1. Oversimplification of causality: The paper appears to oversimplify the concept of causality and its role in prediction tasks, potentially conflating correlation with causation in some instances. In paragraph 1, the example does not clearly demonstrate direct cause-and-effect relationships. For instance, a sporting event doesn’t directly cause increased electricity usage - it’s correlated with increased usage due to more people using electrical devices to watch the event.

2. Lack of precision in terminology: The use of terms like “dissimilar information” in Figure 1 is vague and doesn’t clearly explain what causal information might be captured that similarity-based approaches miss. In addition, the paper seems to imply that causal approaches are superior to similarity-based approaches in all cases, without providing sufficient evidence for this claim.

3. Overgeneralization: The authors make broad claims in their first contribution about the applicability of their approach to “a large class of time series data” without specifying the types of data or providing adequate evidence.

4. Absence of ablation studies: There’s no mention of ablation studies to demonstrate the impact of specific components, such as the residual connections (line 236-238), on the model’s performance and preservation of time step information.

5. Oversimplification of complex systems: The paper seems to underestimate the complexity of systems like weather (Figure 3), suggesting that simple causal relationships can capture their full complexity.  In this example: Some of the described relationships may be indirect. For example, increased cloud cover associated with precipitation could lead to decreased solar radiation and temperature, rather than the precipitation itself directly causing these changes. In addition, weather systems often involve feedback loops where variables influence each other in complex ways. For instance, increased humidity can lead to more precipitation, which in turn affects humidity levels.

6. Insufficient rigor in establishing causality: The paper doesn’t describe a sufficiently rigorous approach to establishing true causality, which would require consideration of potential mechanisms, controlled experiments (where possible), and careful examination of temporal sequences and potential confounding factors.

**Questions:**

1. Causality Understanding:
- How does the paper differentiate between correlation and causation, particularly in the first paragraph's examples?
- Can we truly claim a direct causal relationship between sporting events and electricity usage, or is this more of a correlation due to viewer behavior?
2. Terminology Clarity:
- What specific meaning does "dissimilar information" convey in Figure 1?
- What evidence supports the superiority of causal approaches over similarity-based methods?
- How are these differences quantified and demonstrated?
3. Generalizability Claims:
- What specific types of time series data does this approach effectively handle?
- What evidence supports the claim of applicability to "a large class of time series data"?
- What are the limitations or boundaries of this approach?
4. Model Evaluation:
- How do individual components, particularly the residual connections, contribute to the model's performance?
- What would ablation studies reveal about the preservation of time step information?
- Which components are essential versus optional for model success?
5. System Complexity:
- How does the model account for indirect relationships in weather systems, such as the cloud cover-precipitation-temperature chain?
- How are feedback loops and mutual influences between variables handled?
- Does the model oversimplify complex environmental systems?
6. Causality Validation:
- What methods were used to establish true causality rather than correlation?
- How were potential confounding factors identified and controlled?
- What role did temporal sequences play in establishing causal relationships?
- Were controlled experiments or alternative validation methods considered?

---

### Meta-Review · Area_Chair_F2am · 2024-12-17

**Metareview:**

The paper introduces SPACE (Series Prediction Augmented by Causality Estimation), a novel model for multivariate time series forecasting. SPACE incorporates causal relationships, independent factors, and temporal dependencies to improve forecasting performance. The model integrates a Sequence Enhancer with attention mechanisms, a Cross-TE module to estimate transfer entropy for causal relationships, and a Causal Graph Neural Network (CGNN) that leverages the causality matrix as an adjacency matrix. The authors argue that traditional time series methods fail to capture complex causal relationships, which SPACE aims to address. Experimental evaluations demonstrate improved forecasting performance across multiple real-world datasets.

Strengths

+ The model is motivated by addressing causal structures, which many existing works overlook.
+ Encouraging experimental results, showing SPACE outperforms state-of-the-art methods on nine real-world datasets.
+ Claims improved interpretability of forecasts through learned causal relationships, particularly on weather data.
+ Consideration of model scalability as part of the design.

Weaknesses

+ The causality estimation approach oversimplifies causal relationships and may conflate correlation with causation (e.g., the sporting event-electricity usage example).
+ Strong assumptions regarding pseudo transfer entropy (pTE), such as the normal distribution assumption, may not hold for real-world time series.
+ Limited comparisons with recent and related works, such as TimeMixer, Timexer, and CrossGNN.
+ Lack of ablation studies to validate the contributions of key components like Cross-TE and residual connections.
+ The evaluation datasets are relatively small; larger datasets like NY Taxi or Traffic were not included.
+ The paper lacks efficiency studies (e.g., runtime comparisons with models like iTransformer or DLinear).
+ Unsubstantiated claims in figures (e.g., Figure 1) and unclear evaluations of causal adjacency matrices' advantages.
+ Presentation issues, including inconsistent notation, unclear summation in equations, and redundant or unclear figures (e.g., Figure 2).

**Additional Comments On Reviewer Discussion:**

No rebuttal or discussion.

---

### Decision · Program_Chairs · 2025-01-22

Reject